# The Effects of Two Kinds of Dietary Interventions on Serum Metabolic Profiles in Haemodialysis Patients

**DOI:** 10.3390/biom13050854

**Published:** 2023-05-18

**Authors:** Lucyna Kozlowska, Karolina Jagiello, Krzesimir Ciura, Anita Sosnowska, Rafal Zwiech, Zbigniew Zbrog, Wojciech Wasowicz, Jolanta Gromadzinska

**Affiliations:** 1Laboratory of Human Metabolism Research, Warsaw University of Life Sciences, 02-776 Warsaw, Poland; 2Department of Environmental Chemistry and Radiochemistry, Faculty of Chemistry, University of Gdansk, 80-308 Gdansk, Poland; 3QSAR Lab Ltd., 80-172 Gdansk, Poland; 4Department of Physical Chemistry, Medical University of Gdansk, 80-416 Gdansk, Poland; 5Dialysis Department, Norbert Barlicki Memorial Teaching Hospital No. 1, 90-001 Lodz, Poland; 6B. Braun Avitum Dialysis Center, 93-515 Lodz, Poland; 7Department of Environmental and Biological Monitoring, Nofer Institute of Occupational Medicine, 91-348 Lodz, Poland

**Keywords:** haemodialysis, untargeted metabolomic, dietary intervention, biomarkers of nutrition

## Abstract

The goal of this study was to evaluate the effects of two kinds of 24-week dietary interventions in haemodialysis patients, a traditional nutritional intervention without a meal before dialysis (HG1) and implementation of a nutritional intervention with a meal served just before dialysis (HG2), in terms of analysing the differences in the serum metabolic profiles and finding biomarkers of dietary efficacy. These studies were performed in two homogenous groups of patients (n = 35 in both groups). Among the metabolites with the highest statistical significance between HG1 and HG2 after the end of the study, 21 substances were putatively annotated, which had potential significance in both of the most relevant metabolic pathways and those related to diet. After the 24 weeks of the dietary intervention, the main differences between the metabolomic profiles in the HG2 vs. HG1 groups were related to the higher signal intensities from amino acid metabolites: indole-3-carboxaldehyde, 5-(hydroxymethyl-2-furoyl)glycine, homocitrulline, 4-(glutamylamino)butanoate, tryptophol, gamma-glutamylthreonine, and isovalerylglycine. These metabolites are intermediates in the metabolic pathways of the necessary amino acids (Trp, Tyr, Phe, Leu, Ile, Val, Liz, and amino acids of the urea cycle) and are also diet-related intermediates (4-guanidinobutanoic acid, indole-3-carboxyaldehyde, homocitrulline, and isovalerylglycine).

## 1. Introduction

Nutritional factors are stronger predictors of short-term and long-term outcomes and mortality in haemodialysis (HD) patients. In those HD patients with low glomerular filtration rates, the metabolic waste products are insufficiently excreted from the body [1]. Approximately 90 substances found in the blood of HD patients are urotoxic [2]. Several studies have demonstrated the catabolic effects of HD procedures, which have been explained by the loss of amino acids and proteins in the dialysate [3,4] and HD-induced inflammatory responses [5]. In HD patients, a decrease in nutritional status is typically observed after a transient improvement in nutritional status during the first six months of treatment [6]. This may be due to decreased protein and energy intake caused by anorexia, dietary restrictions due to the need to reduce the consumption of selected minerals, metabolic alterations in the organs involved in nutrient digestion and absorption (intake) and the inability to obtain or prepare food. Poorer nutritional status impairs physical activity, increases morbidity and mortality and is also related to cardiovascular incidences, hospitalisations, low quality of life and poor prognoses for HD patients [7]. In HD patients, a 1 g/dL decrease in serum albumin is associated with an increased mortality risk of 47% compared to 38% in patients treated with peritoneal dialysis. Persistent depletion of proteins and energy in the body due to poor nutritional status is caused by multiple nutritional and non-nutritional mechanisms, as well as by metabolic alterations [8]. The nutrient loss into dialysate and hypercatabolism associated with dialysis-related inflammation and oxidative stress also contribute to the severity of protein–energy malnutrition [9,10,11,12,13]. Inflammation associated with HD has a direct impact on increased protein catabolism by activating the ubiquitin–proteasome system, caspase-3 and myostatin [14]. The inhibition of inflammation has a beneficial impact on improving the nutritional status of patients treated with HD [15].

The assessment of the dietary habits and nutritional status of HD patients performed by the Institute of Food and Nutrition in Warsaw showed lower energy values, by as much as 30%, compared to the European Guidelines for Nutritional Care of Renal Adult Patients [16]. The analysis of basic nutrient intake indicated a lack of a balanced content of essential nutrients in daily food ratios: too low protein and carbohydrate intake in women and dietary fibre in both men and women were observed [17]. An important factor contributing to the loss of lean body mass is lower energy and nutritive values in the diet on days with dialysis as compared to days in which this procedure does not take place. Laville and Fougue described that HD patients consumed only 80% of the daily food rations. This means that their energetic deficiency was 2800 kcal per week [18]. The main reasons for missing the meals were repeated laboratory tests after an overnight fast, modifications to the dialysis timetable, tiredness after dialysis, and a restricted diet not adapted to individual dietary habits. Getting to the dialysis centre takes a long time for patients, and the dialysis procedure lasts approximately 4 h. All of this means that patients on days with dialysis consume much less food [19]. 

It has also been shown that the administration of enteral or parenteral nutrition during a dialysis procedure can compensate for the catabolic effects of HD [20,21]. Protein metabolism studies using stable isotopes have demonstrated that the introduction of intradialytic parenteral nutrition (IDPN) or enteral nutrition (EN) during dialysis would maintain a positive metabolic effect in both skeletal muscle and throughout the body. Veeneman et al. [20] examined the effects of consumption of six liquid portions (yoghurt, cream, and milk protein powder) every 30 min on whole body protein metabolism (method with primed constant infusion of L-[1-13C]valine) in HD patients on a day without dialysis and during dialysis. According to the study protocol, the patients consumed about 50% of their daily protein requirements. It was shown that, on a non-dialysis day, the protein balance was negative after an overnight fast. Consumption of the above-mentioned portions resulted in a positive whole-body protein balance. During dialysis, consumption of the same portions resulted in a positive protein balance to the same degree as on a non-dialysis day. In the opinion of the authors, this amount of protein might be excessive for HD patients during a 4-h dialysis treatment. In another study on protein metabolism during and after HD procedures, IDPN and EN were used with oral supplementation of industrial, ready-to-eat liquid diets. IDPN was started 30 minutes after HD initiation and continued until the end of dialysis. EN was administered as three equal meals: 0.5 h, 1.5 h, and 2.5 h after the start of dialysis. In this study, the serum insulin and amino acid concentrations were analysed both during dialysis and in the post-dialysis period. During dialysis, both for EN and IDPN, serum amino acids increased simultaneously with insulin. The authors suggest that these effects are probably the result of increased serum amino acids from the administered meals and an increased serum insulin level. It is known that insulin not only influences carbohydrate homeostasis but also regulates protein metabolism by stimulating amino acid transport, inhibiting proteolysis, and promoting protein synthesis [21]. In-depth information on why an extra meal versus a missing meal is more beneficial is also included in our first paper [22].

Currently, there is no simple method of single biomarker analysis that would describe the nutritional status of HD patients. Several biomarkers are analysed in HD patients to assess their nutritional status, and most often they include the serum prealbumin, albumin, transferrin, creatinine, and cholesterol concentrations [10]. No direct evidence has been evaluated concerning the metabolic response to the HD procedure with a meal administered immediately before the HD.

Summing up the current state of knowledge, the scientific goal of this study was to evaluate the effects of two kinds of long-term dietary interventions, the implementation of a nutritional intervention with a meal served just before dialysis and a traditional nutritional intervention without a meal before dialysis, in terms of improving the nutritional status of HD patients, analyse the differences in the serum metabolic profiles after the implementation of these two kinds of nutritional interventions and identify potential biomarkers of the efficiency of the nutritional interventions between the HD groups.

## 2. Materials and Methods

### 2.1. Patients

The study involved 70 patients treated at 2 dialysis centres located in Lodz. The patients included in the study were clinically stable (men and women) with HD for at least 6 months before the study. The patients underwent a standard haemodialysis procedure 3 times a week for 180 to 300 min, depending on the effective blood flow. Low-flux-type dialysers were used, and the area of the dialysis membrane was 1.6 to 2.2 m^2^. Patients were excluded from the study if they had active inflammatory and infectious diseases, a history of malignancy in the past 5 years, chronic liver disease (Child–Pugh C), diabetes treated with insulin, or were taking glucocorticoids and nonsteroidal anti-inflammatory drugs. The enrolled patients were divided into 2 groups:

Group HG1—patients with the dietary recommendations without a meal served before the dialysis (n = 35) for 24 weeks.

Group HG2—patients with the dietary recommendations with a meal served just before the dialysis (n = 35) for 24 weeks.

At baseline, the patients in HG1 and HG2 received dietary recommendations based on nutritional exchanges with individually calculated energy and nutrient values, together with a plan for meals on the days with dialysis and the days without dialysis [21].

The European Best Practice Guidelines [23] were incorporated into developing the personalised diet plans. For the patients in HG2, just before the dialysis procedures, they were served meals prepared by a diet catering company. The nutritional value of these meals was calculated as 20% of the mean energy and nutrient requirements: energy 416 ± 23 kcal, protein 16.4 ± 0.6 g, fat 14.2 ± 1.6 g, total carbohydrates 58.2 ± 5.6, potassium 452.7 ± 79.6 mg, and phosphorus 183.9 ± 10.5 mg. The meals were delivered to the dialysis centre every day. The products were high-quality, tasty, and had a well-defined weight (e.g., Basque trout, rice, salad with colourful lettuce and cranberries; chicken with tarragon sauce, pasta, and carrots). From the diet were excluded foods that were difficult to digest and could lead to the occurrence of gastrointestinal disturbances. The energy and nutrient values of the served meals were incorporated into the daily energy and nutrient demands. This meant that the recommended nutritional values on days with and without dialysis were the same. This also meant that in both HG1 and HG2, the nutritional value of the daily food rations was calculated in the same way. Detailed data on the dietary recommendations, assessments of nutrition, and nutritional status are included in our previous publication [22,24].

The study protocol was approved by the Local Ethical Committee at the Nofer Institute of Occupational Medicine in Lodz (protocol no. 08/201 30 May 2018), and all the patients gave their written informed consent for participation in the study. All the patients were studied at baseline and after 24 weeks of the nutritional intervention. Blood samples from the patients in HG1 and HG2 (7.5 mL) were collected during the midweek dialysis into S-Monovette^®^ test tubes (Sarstedt AG 7KG, Numbrecht, Germany) with a cloth activator twice: before (t = 0) and after 24 weeks (t = 24) of the nutritional intervention (always before dialysis). The serum was separated via centrifuging (10 min, 2000 rpm) and frozen afterwards in Eppendorf tubes at −80 °C until the metabolomic profiling and biochemical determinations.

### 2.2. Analytical Methods

In the beginning and after 24 weeks of the study, prior to the dialysis session, the blood morphology and biochemical parameters were determined in the serum: creatinine, urea, Ca, P, Na, K, albumin, total cholesterol, and transferrin. The laboratory blood parameters (blood morphology, creatinine, urea, albumin, total cholesterol, Ca, P, K, and Na) were checked using standard laboratory techniques. The concentrations of transferrin were analysed using ELISA tests: Human Transferrin ELISA Kit (Bioassay Technology Laboratory, Yangpu, China).

Nutrition was assessed using 3 days of dietary records before and after the 24 weeks of nutritional intervention. This part of the study was performed by qualified personnel. A computer program, Dieta 5 (Warsaw, Poland), was used to determine the daily energy and nutrient intake.

### 2.3. Serum Sample Preparations and Analysis of Untargeted Metabolomics

Materials: Samples of human plasma were split into 80 μL aliquots and frozen at −80 °C until preparation. The acetonitrile (ACN), methanol (MeOH) and chloroform (CHCl3) were all LC–MS grade (Avantor Performance Materials Poland S.A., Gliwice, Poland), ammonium formate 99.99% Sigma-Aldrich (St. Louis, MO, USA), benzoyl-D5 98% and L-phenylalanine 3,3-D2 98% Cambridge Isotope Laboratories, Inc. (Andover, MA, USA), and CUDA (formal name 12-[[(cyclohexylamino)carbonyl]amino]-dodecanoic acid) Cayman Chemical Company (Ann Arbor, Michigan, USA). The ultra-high-purity water was prepared with an R5 UV Hydrolab system (Wislina, Poland). The sodium formate calibration solution and leucine encephalin lock mass solution (Waters, UK) were prepared according to the manufacturer’s specifications.

The serum sample extraction and analysis were carried out on the basis of a protocol developed by Southam et al. [25]. For the CHCl3 method, 512 μL of ice-cold methanol, 125 μL of ice-cold water, and 7 μL of mixture internal standards (benzoyl-D5 and L-phenylalanine 3,3-D2) were added to plasma (80 μL, thawed on ice) and vortexed (60 s). Then, 512 μL of ice-cold chloroform and 256 μL of ice-cold water were added and the samples were vortexed (60 s), incubated (10 min in a Thermomixer, Eppendorf, Hamburg, Germany), and centrifuged (21,000× *g*, 10 min, 4 °C). The samples were set at room temperature (approximately 20 °C) for 20 min to allow for biphasic partitioning, and 70% of the polar phase (600 μL) was removed and dried in a SpeedVac sample concentrator (Eppendorf, Hamburg, Germany). All the polar extracts were resuspended in 80 μL of mixture (59 μL of ACN, 20 μL of H_2_O, and 1 μL of CUDA), vortexed (30 s), centrifuged (21,000× *g*, 20 min, 4 °C) and 60 μL of the supernatant was aliquoted into a low recovery volume HPLC vial.

Metabolomic profiling, including polar and semi-polar metabolites, was performed using a Waters Acquity^TM^ Ultra Performance LC system (Waters Corp., Milford, MA, USA) connected to a Synapt G2Si Q-TOF (Quadrupole Time-of-Flight) mass spectrometer (Waters MS Technologies, Manchester, UK) equipped with an electrospray source (ESI) (Waters, Manchester, UK). An ACQUITY UPLC BEH Amide Column (130 Å, 1.7 µm, 2.1 mm × 100 mm) was applied with an ACQUITY UPLC BEH HILIC 1.7 um VanGuard Pre-Column (Waters Corporation, Milford, MA, USA). The injection volume of the serum extract in the ESI^+^ was 6 μL and in the ESI^−^ was 10 μL. The gradient mobile phase was a mixture of acetonitrile, ammonium formate (1 M), and formic acid, and the water for mobile phase A was 95% acetonitrile/water (10 mM ammonium formate, 0.1% formic acid), while for mobile phase B it was 50% acetonitrile/water (10 mM ammonium formate, 0.1% formic acid). Data were acquired in the positive and negative ionisation modes separately. The gradient elution for the ESI^+^ was performed in the following manner: 0.0–1.5 min 1% phase B, 1.5–2.0 min from 1% to 15% phase B, 2.0–6.0 min from 15% to 50% phase B, 6.0–9.0 min from 50% to 95% phase B, 9.0–10.0 min from 95 to 99% phase B, 10.0–10.5 min from 99% to 1%, and 10.5–14.0 min 1%. The gradient elution for the ESI^−^ was as follows: 0.0–2.5 min 1% phase B, 2.5–3.5 min from 1% to 10 % B, 3.5–6.0 min from 10% to 50% B, 6.0–7.5 min from 50% to 95% B, 7.5–8.5 min from 95% to 99%, 8.5–10.0 min 99%, 10.0–10.5 min from 99% to 1% and 10.5–13.0 min 1%. Nitrogen was used as the cone and desolvation gas. The desolvation gas flow was set at 900 L/h, cone gas flow was set at 50 L/h, and the source temperature was 120 °C. The capillary voltage for the ESI^+^ ion mode was 3.5 kV, and for the ESI^−^, it was 2.7 kV. All the data were acquired using the lock mass to ensure accuracy and reproducibility. Leucine enkephalin was used as the lock mass at a scan time of 0.5 s, interval of 15 s, and mass window of 0.5 Da. The MS method was used for the data collection, with a scan time of 0.5 s, centroid data format, and high-resolution mode.

Quality control (QC) samples were prepared by pooling equal volumes of extracted serum from each sample. The QC was injected at the beginning of the run and every 10th sample to monitor the system consistency. Potentially identified compounds obtained from the data processing were subjected to fragmentation using the collision energy of 20 V in MS/MS mode. The results of the compound fragmentation were compared to the spectra in the Human Metabolome Database (HMDB) [26].

### 2.4. Statistical Analysis

The Shapiro–Wilk W test was used to evaluate the normal distribution of the continuous data and expressed as the mean ± SD. Differences between the continuous variables before and after the 24 weeks of nutritional intervention were assessed using the ANOVA and post-hoc NIR Fisher tests.

A *p* < 0.05 was accepted as statistically significant. The statistical analyses were performed using Statistica v. 13.1 software (StatSoft Inc., Tulsa, OK, USA). A statistical and chemometric analysis univariate statistical analysis was performed using the *stats models* Python module.

The Student’s *t*-test or Mann–Whitney U test was applied after normality checking for the independent samples, whereas the Wilcoxon test was used to analyse the paired samples.

Next, the statistically significant metabolites with the largest fold changes were selected using volcano plots. These scatter plots visualised the 2D space created by the log2 of the fold changes (log2FC, x-axis) and -log10 of the p-value (y-axis). Metabolites that did not show differences at the assumed level of statical significance were marked in grey. In contrast, statistically significant metabolites with a relatively low fold change (|log2FC| > 0.10) were marked in blue. The statistically substantial downregulated and upregulated metabolites were marked in red and green, respectively. A principal component analysis (PCA) was performed with the *R* (version 4.2.0, https://www.R-project.org/, accessed on 14 May 2023) environment using the "factoextra" and "FactoMineR" libraries. The metabolomic data matrices were analysed before and after annotating the most important metabolites.

## 3. Results

The study was performed in two groups of patients with chronic kidney disease treated with HD. The HG1 group received individually prepared dietary recommendations, which were customised for days with dialysis and for days without dialysis. The HG2 group received a specially prepared meal immediately before the start of dialysis in addition to the individually prepared recommendations. The studied group was composed of 51–57% men, and no observed significant differences were observed between the two groups with regard to age, vintage, time of professional work, smoking habits, medical treatment, or supplements used (Table 1).

The patients in the HG2 group had statistically significantly higher serum urea and creatinine concentrations than the patients in HG1 at the beginning of the study. These differences remained statistically significant even after the end of the study (Table 2). On the basis of the dietary intake questionnaires, the energy value of the diet and the daily intake of basic food products were assessed. The parameters assessing the diet of the patient groups participating in the study were statistically insignificant per kg bw.

After 24 weeks of following the dietary recommendations plus a meal before dialysis, there were no statistically significant changes in body weight or BMI from baseline in the two study groups. Among the analysed biochemical parameters, there was a statistically significant increase in albumin and transferrin in HG2 compared to HG1 after the end of the study (Table 2).

### UPLC Q-TOF Peak Signal Intensity

In the metabolomic profile, a total of 7430 metabolite peaks were detected. On the basis of the UPLC Q-TOF/MS peak intensity chromatograms of HG2 at the start of the study (HG2 t = 0) versus after 24 weeks of the study (HG2 t = 24), and the HG1 group vs. the HG2 group after the end of the study (HG1 t = 24 vs. HG2 t = 24), see the volcano plots presented in Figure 1A,B. Among all the metabolite peak intensities on the chromatograms, it was found that 180 metabolites were statistically different between the groups HG1 vs. HG2 at t = 24 and 97 metabolites differed between HG2 at t = 0 and HG2 at t = 24. The statistical significances between the identified peak intensities of both investigated groups after 24 weeks of dietary intervention are presented in Table 3. Among the metabolites with the highest statistical significance between HG1 and HG2 after the end of the study, 21 substances were putatively annotated, which had potential significance in both of the most relevant metabolic pathways in the cell and those related to diet (Table 3).

1—6-hydroxymethyl-7,8-dihydropterin, 2—4-hydroxyphenylpyruvic acid, 3—benzenesulphonic acid, 4—5-hydroxyvalproic acid, 5—p-coumaric acid sulphate, 6—4-(glutamylamino)butanoate, 7—homocitrulline, 8—2,6-diamino-4-hydroxy-5-N-methylformamidopyrimidine, 9—6,8-dihydroxypurine, 10—indole-3-carboxaldehyde, 11—diethyleneglycoldimethacrylate, 12—isovalerylglycine, 13—5-hydroxyindoleacetylglycine, 14—3-(aminomethyl)-5-methylhexanoic acid, 15—5-(hydroxymethyl-2-furoyl)glycine, 16—4-guanidinobutanoic acid, 17—norepinephrinesulphate, 18—gamma-glutamylthreonine, 19—N(4)-acetylsulphisoxazole, 20—metanephrine, 21—tryptophol.

The results of the PCA analysis performed on the matrix, including all the metabolites, indicated that the groups of patients, with and without additional meals, were not separated (Figure 2).

Importantly, the complete separation of these groups is achieved based on the metabolites selected using the volcano plots. In Figure 3, the space created by the score of the first principal component (Dim_1 –_ dimension 1 = variable grouped in the horizontal direction) distinguishes patients with and without additional meals. After the 24 weeks of dietary intervention, the main differences between the metabolomic profiles in the HG2 vs. HG1 groups were related to the higher signal intensities from amino acids and protein metabolites: indole-3-carboxaldehyde, 5-(hydroxymethyl-2-furoyl)glycine, homocitrulline, 4-(glutamylamino)butanoate, tryptophol, gamma-glutamylthreonine, isovalerylglycine and others.

These metabolites are intermediates in the metabolic pathways of the following amino acids: Trp, Tyr, Phe, Leu, Ile, Val, Liz, and amino acids of the urea cycle. Another group of metabolites comprises protein or amino acid breakdown products, which are at the same time breakdown products of the animal proteins supplied with the food: hydroxyindoleacetylglycine, homocitrulline, 5-(hydroxymethyl-2furoyl)glycine, and gamma-glutamylthreonine. Some intermediates of the catabolic amino acid metabolism occurring in the cells may also be markers of selected plant components of the diet. Indole-3-carboxaldehyde is determined after the consumption of beans, cucumbers, cereals, and cereal products, while 4-guanidinobutanoic acid is determined after the consumption of certain types of fruits. The significantly different metabolites in the study groups of HD patients include lipid-related compounds: 5-hydroxyvalproic acid, isovalerylglycine, representing the metabolic pathways of fatty acids, and cholesterol. Among the putatively annotated metabolites, the study demonstrated markers of environmental and occupational exposure: N(4)-acetylsulphisoxazole, diethyleneglycoldimethacrylate, 3-(aminomethyl)-5-methylhexanoic acid, and benzenesulphonic acid.

Comparing the metabolic markers of the HG2 group before and after the start of the study, a 4.7-fold increase in signal intensity was found for gamma-glutamylthreonine. When comparing the signal intensity between the groups of HD patients (HG1 vs. HG2 at t = 24), the greatest differences were revealed for metabolites of natural amino acid metabolism, which are also diet-related intermediates: 4-guanidinobutanoic acid, indole-3-carboxyaldehyde, L-homocitrulline, and isovalerylglycine (Figure 4).

## 4. Discussion

HD is a complex procedure that removes toxic metabolites and excess water from the body, thus changing the metabolic status of the patient. Treating chronic kidney disease (CKD) patients with HD is associated with the loss of low-molecular-weight components, including amino acids. During one haemodialysis cycle, approximately 12 g of amino acids are lost, accompanied by the proteolysis of muscle proteins [27].

Dietary restrictions traditionally implemented in patients with CKD, especially with end-stage CKD, may influence not only the metabolome but also the microbiome of such patients [28]. Metabolomic profiling is influenced by genetic variations, liver function, medications administered, concomitant chronic diseases and, most importantly, the magnitude of the estimated glomerular filtration rate [29]. Differences in amino acid metabolism are clearly visible as a consequence of the introduction of meals during HD. Metabolomic analysis of both groups of HD patients after 24 weeks of dietary intervention identified 181 compounds whose signal intensity in the metabolomic analysis differed statistically between the two groups. Among these significantly different metabolites between the HG1 and HG2 groups, amino acid metabolites and low-molecular-weight breakdown products of proteins, amino acids, and fatty acids, as well as purine and pyrimidine metabolites and substances that are metabolites of both plant and animal dietary components, were identified based on the HMDB. After the dietary intervention, the main differences between the metabolomic profiles in the HG2 vs. HG1 groups were connected with higher signal intensities from indole-3-carboxaldehyde, homocitrulline, 4-(glutamylamino) butanoate, 5-hydroxyindoleacetylglycine, metanephrine, 5-(hydroxymethyl-2-furoyl)glycine, tryptophol, 2,6-diamino-4-hydroxy-5-N-methylformamidopyrimidine, norepinephrine sulphate, gamma-glutamylthreonine, and isovalerylglycine. 

Isovalerylglycine is an intermediate of the catabolic leucine (Leu) pathway. The catabolism of essential branched-chain amino acids (Leu, Val, and Ile) is sensitive to nutritional alterations. Isovaleric acid is a normal metabolite of Leu. The first step in the degradation pathways of branched-chain amino acids is catalysed by the aminotransferase to form the corresponding branched-chain keto-acids. The next Leu in oxidative decarboxylation contributes to the generation of acyl-CoA derivatives, in this case isovaleryl-CoA [30]. This compound is formed in the case of a metabolic block caused by mitochondrial isovaleryl-CoA dehydrogenase deficiency. There is then an accumulation of isovaleryl-CoA metabolites in the body.

Isovaleryl acid is also a quantitatively less important metabolite in disorders of mitochondrial fatty acid β-oxidation. The accumulation of isovaleric acid indicates a defect in the catabolism of short-chain fatty acids [31]. Defects in fatty acid oxidation affect tubular epithelial cells, causing the accumulation of fatty acids in these cells [29]. Conjugation metabolites of fatty acids with glycine serve an important detoxification role [32].

The occurrence of homocitrulline in the plasma accompanies an increase in the ornithine (Orn) and ammonia concentrations. This compound with an alpha-amino acid structure is formed by the non-enzymatic carbamoylation of the epsilon-amino group of lysine (Lys). The carbamoylation process occurs in humans in two cases: in uraemia and inflammation. Homocitrulline is proposed as a potential biomarker in various pathological states involving urea cycle disorders [29,33]. As shown by Jaisson et al., the homocitrulline concentration decreases by approx. 50% during the first 6 months of HD, and it remains at the same level afterwards. Studies conducted by these authors have shown that homocitrulline in the serum occurs in free and protein-bound forms, with approximately 90% of the metabolite occurring in the bound form [34]. Of note, it is likely that the bound form of homocitrulline is responsible for the stable concentration of the metabolite. In the case of residual diuresis, there is a decrease in the homocitrulline concentration due to catabolism and the elimination of low-molecular-weight proteins. The above information refers to the role of homocitrulline in uraemia. At the same time, this compound is a product of the catabolic metabolism of animal proteins supplied with food. 

Among the identified metabolites, two compounds formed by the arginine (Arg) metabolism cycle were described: the butanoic acid derivatives 4-guanidinobutanoic acid and 4-(glutamylamino)butanoic acid. 4-(glutamylamino)butanoate is a polyamine that is an intermediate in the degradation of putrescine, a toxic biogenic amine formed during metabolism catalysed by microorganisms from arginine and ornithine. At high concentrations, this compound may be involved in protein synthesis inhibition processes. 4-guanidinobutanoic acid belongs to the gamma group of amino acids and to the uremic toxins. In normally functioning kidneys, this metabolite is normally excreted from the kidneys [35].

4-guanidinobutanoic acid has been detected but not quantified in several different foods, such as apples and exotic fruits (e.g., loquats). This could make 4-guanidinobutanoic acid a potential biomarker of the consumption of these foods. 4-hydroxyphenylpyruvic acid is an intermediate in the phenylalanine (Phe) and tyrosine (Tyr) catabolic pathways [36]. 4-hydroxyphenylpyruvic acid is a metabolite of tyrosine aminotransferase and 4-hydroxyphenylpyruvic acid dioxygenase. This substance is impossible to determine in the plasma of a healthy human. The deficiency of this enzyme or the inhibition of its activity, which in HD patients may be caused by the presence of toxic metabolites, may cause the accumulation of 4-hydroxyphenylpyruvic acid in the serum or tissues. The concentration of 4-hydroxyphenylpyruvic acid does not affect kidney functions, although it increases with the degree of inflammation, including uraemia [37].

The metabolic pathways of Phe are directly affected by the availability of the renal and hepatic enzyme phenylalanine 4-hydroxylase. This enzyme converts Phe into Tyr, while the Duranton et al. study found the tendency to decrease the Tyr concentration in CKD patients [37].

The same concerns metanephrine and norepinephrine sulphate. These are compounds classified as catecholamines, which are formed as products of Tyr and Phe metabolism in the subway of adrenaline and noradrenaline degradation. The metamephrine concentration was found to increase in 25–28% of HD patients and was responsible for hypertension in this group [38].

Gamma-glutamylthreonine is a product of incomplete protein digestion or catabolism, a short-lived intermediate, and subject to further proteolysis. Bernard et al. used Netboost in the training dataset to form 43 clusters of metabolites, with Cluster 5 containing 5-oxoproline and its metabolites, gamma-glutamyl amino acids, which were associated with a lower risk of CKD and the end-stages of CKD [39]. It is also believed that gamma-glutamylthreonine may be a marker of the presence of sugars in the diet [40].

The other identified compounds were metabolites in the catabolic tryptophan (Trp) pathway, tryptophol and indole-3-carboxaldehyde. Both Trp metabolites are indole derivatives, metabolites derived from the degradation of the amino acid by the gut microbiota [41].

Among the identified diet-related metabolites, an important role is played by the products of tryptophan catabolism. Tryptophan is an exogenous amino acid, a precursor to the neurobiologically important compounds serotonin and melatonin. At the same time, the UroSystemOmics study showed that the Trp concentrations in patients with stage 5 CKD (independent of the underlying aetiology) are approximately 50% lower than in patients with stage 3. Approximately 75–90% of circulating tryptophan occurs in a bound form, primarily with albumin; however, the rate of the Trp decrease is significantly faster than that of albumin in CKD patients [39]. Thus, hypoalbuminemia is not the cause of low Trp. In terms of its chemical structure, tryptophol is an indoxyl alcohol. At the same time, indole-3-caboxaldehyde is a marker of dietary ingredients such as beans, cucumbers, *Cruciferous* vegetables, cereals and cereal products.

In terms of its chemical structure, hydroxyindolacetylglycine is an n-acyl-alpha-amino acid formed by the metabolic conversion of Trp and the branched-chain amino acids Val, Leu, and Ile. Detected in the serum, it can also be a marker of poultry meat consumption.

In HD patients, quantitative and qualitative alterations in plasma amino acids and their metabolite profiles have been observed. Dietary intervention improves the nutritional status of HD patients, although progress in CKD would affect amino acids and their metabolite profiles. The kidney plays an important role in the metabolism of proteins and amino acids. In the above study, it was found that there were statistically significant changes in the signal intensity, mostly between the essential amino acid metabolites. The significance of quantification between amino acid metabolites is not easy to explain. The catabolic pathways of amino acids are affected by malnutrition (Val/Gly ratio), inflammation and acidosis, renal enzyme activity modified by kidney disease (glycine hydroxymethyltransferase: Gly→Ser; phenylalanine 4-hydroxylase: Phe→Tyr) or intestinal microbiota (Trp) [26,35]. The catabolic pathways of essential amino acids are mainly multistep pathways. Non-essential amino acids during degradation give small molecules, e.g., two carbon metabolites, which are immediately involved in the citric acid cycle, beta-oxidation, or the synthesis of new amino acids. Analysis of amino acid metabolites is not specific to kidney disease, and it is not clearly defined if they are markers of dietary intervention. 

Among the identified metabolites, a separate group includes compounds that are not natural metabolites of cell metabolism but are identified as metabolites of certain dietary components: 6-hydroxymethyl-7,8-dihydropterin, p-coumaric acid sulphate, and 5-(hydroxymethyl-2-furoyl)glycine. This group of compounds includes 6-hydroxymethyl-7,8-dihydropterin, a metabolite associated with the consumption of certain vegetables and herbs, and p-coumaric acid sulphate detected in body fluids after consumption of products prepared or manufactured from whole grains. Some intermediates of catabolic amino acid metabolism in cells can also be markers of certain dietary components. Indole-3-carboxaldehyde is detected after the consumption of beans, cucumber, cereals and cereal products, 5-hydroxyindoleacetylglycine is detected in poultry meat, and 4-guanidinobutanoic acid is detected after the consumption of certain types of fruits.

Among the identified metabolites, markers of environmental and occupational exposure were also observed, i.e., N(4)-acetylsulphisoxazole, diethyleneglycoldimethacrylate, 3-(aminomethyl)-5-methylhexanoic acid, and benzenesulphonic acid.

## 5. Conclusions

The results presented in this study reflect a preliminary attempt to analyse the effects of providing dialysis patients with a meal immediately prior to the dialysis procedure.

An undeniable effect of providing the meal is the increase in the serum albumin and transferrin concentrations and the appearance of essential amino acid metabolites after 24 weeks of dietary intervention compared to the group of patients where only the dietary recommendations were applied.

Another effect concerns the metabolomic profile of the compounds found in the serum: the quantitative distribution of metabolites of natural metabolic processes varies, and the metabolites of substances formed during the metabolic pathways of consumed food products also appear.

While the presented results indicate a broad spectrum of potential biomarkers that may be useful in determining the nutritional status of HD patients, several limitations of this study should be considered. The studied patients did not form a homogeneous group, as their age, disease duration, and urea and creatinine serum concentrations, as well as probably their genetic profiles, varied significantly. The metabolic profile was also influenced by the comorbidities experienced and the medications used, as the patients were from two different medical centres. The patients in this study were not rigorously categorised according to their cause of end-stage CKD or the procedure used for the treatment of other chronic diseases. During this study, we evaluated two groups of HD patients. Probably it would be more accurate to compare the same group of HD patients: first without meals before HD for 24 weeks and then, for the next 24 weeks, with meals consumed before HD. The meals, in the case of the HG2 patients, were always given before the start of dialysis, although due to the different appetites and rates of satiation, it was an incidental case that the patients did not always consume a whole meal before the HD procedure. Metabolomic profiling was only performed at the start and end of the study. Metabolomic profiling was performed using the untargeted metabolomics method, and not all the statistically significant features were putatively annotated.

Despite the limitations, this study had a number of strengths. The dietary intervention during dialysis improved some biochemical markers of the nutritional status of HD patients.

More specifically, it significantly increased the albumin and transferrin concentrations compared to the baseline values. Some significant differences were found in the metabolomic profile expressed as the signal intensity between HG1 and HG2 after 24 weeks of the dietary intervention.

The potential biomarkers of the effectiveness of the dietary intervention were associated with amino acid metabolism (Trp, Arg, Tyr, and Phe), fatty acid pathways, and pyrimidine catabolism.

## Figures and Tables

**Figure 1 biomolecules-13-00854-f001:**
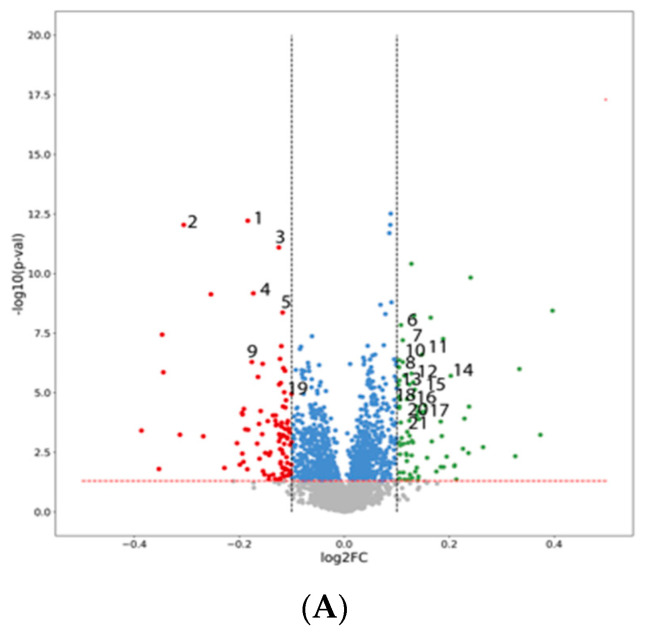
Volcano plots of metabolic data. The x-axis is a mean log2 ratio fold change in the relative intensity of each metabolite between two samples: (**A**) HG1 t = 24 vs. HG2 t = 24 with marked putatively annotated metabolites; (**B**) HG2 t = 0 vs. HG2 t = 24 (grey dotted lines indicate metabolite with |log2FC| > 0.10). The y-axis represents the statistical significance of the *p*-values for each metabolite. The dashed red line indicates the *p*-value equal to 0.05. Metabolites that did not show differences at the assumed level of statistical significance were marked in grey; statistically significant metabolites with a low fold change (|log2FC| > 0.10) were characterized in blue, statistically significant downregulated metabolites were marked in red and statistically significant upregulated metabolites were marked in green.

**Figure 2 biomolecules-13-00854-f002:**
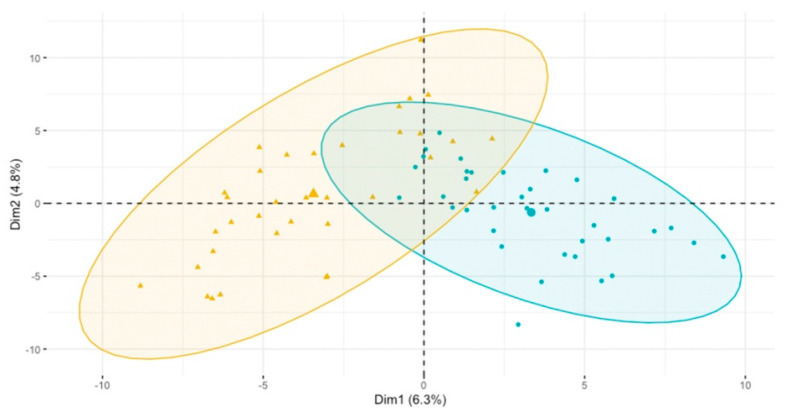
PCA score plot of HG1 t = 24 vs. HG2 t = 24 based on all the investigated metabolites. Blue circles represent the HD1 group, while the yellow symbolize the HD2 group.

**Figure 3 biomolecules-13-00854-f003:**
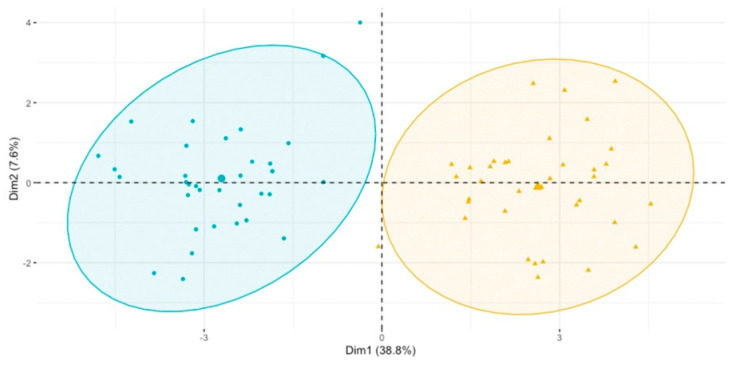
PCA score plot of HG1 t = 24 vs. HG2 t = 24 based on the putatively annotated metabolites. Blue circles represent the HD1 group, while the yellow symbolize the HD2 group.

**Figure 4 biomolecules-13-00854-f004:**
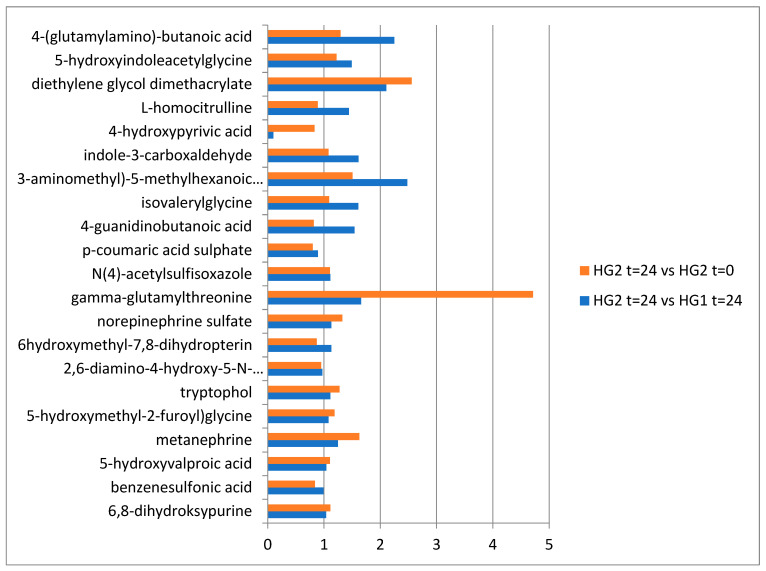
Mean fold changes in relation to the signal intensities from the putatively annotated compounds in the serum of HD patients after 24 weeks of dietary recommendations versus dietary recommendations with an additional meal served (HG2 t = 24 vs. HG1 t = 24) and in the serum of HD patients at the beginning of the study and after 24 weeks of the dietary recommendation with an additional meal served (HG2 t = 0 vs. HG2 t = 24).

**Table 1 biomolecules-13-00854-t001:** Characteristics of the study participants (mean ± SD or n, %).

Variables	HG1, n = 35	HG2, n = 35
Male, n Female, n	21 (57.2%) 14 (42.8%)	18 (51.4%) 17 (48.6%)
Professional work, yrs	5–43	6–40
Age, yrs	59.7 ± 11.4 (32–75)	58.1 ± 15.7 (24–88)
Vintage HD, m	56 (9–145)	54 (6–156)
Smoking, YES/NO Years of smoking Cigarettes/day	11/24 (31.4/68.6%) 18–50 3–20	6/29 (17.1/82.9%) 10–55 5–20
Passive smoking, YES/NO	5/30 (14.3/85.7%)	10/25 (28.6/71.4%)
Medical treatment	35 (100%)	35 (100%)
Vitamin/microelement supplements	14 (40.0%)	17 (48.6%)

**Table 2 biomolecules-13-00854-t002:** Biochemical markers and selected parameters of the diet in the HG1 and HG2 patients before the start of the study (t = 0) and after 24 weeks of dietary intervention (t = 24). Data are presented as values from dialysis days.

Variables	HG1 t = 0 Mean ± SD (Range)	HG2 t = 0 Mean ± SD (Range)	HG1 t = 24 Mean ± SD (Range)	HG2 t = 24 Mean ± SD (Range)
Body mas, kg	74.9 ± 19.4 (36.9–105.6)	71.9 ± 13.2 (49.2–113.2)	75.7 ± 19.5 (39.4–106.0)	71.6 ± 13.0 (49.3–109.7)
BMI, kg/m^2^	26.6 ± 4.9 (15.4–34.3)	25.2 ± 4.9 * (18.7–40.6)	27.1 ± 4.8 (16.4–34.0)	25.5 ± 4.2 (19.8–37.4)
eGFR, mL/(min × 1.73 m^2^)	8.5 ± 5.3 (3.0–27.0)	5.8 ± 1.9 (3.0–13.0)	7.2 ± 2.9 (2.0–18.0)	6.1 ± 3.4 (3.0–22.0)
Energy intake Kcal/day/kg bw	23.4 ± 5.6 (7.58–51.0)	23.4 ± 8.4 (5.70–40.5)	24.7 ± 5.0 (9.0–42.4)	27.1 ± 6.4 ** (11.5–35.2)
Protein intake, g/day/kg bw	1.01 ± 0.26 (0.35–2.73)	1.04 ± 0.45 (0.23–1.91)	1.11 ± 0.32 (0.37–2.13)	1.19 ± 0.30 (0.44–1.70)
Urea, g/L	0.123 ± 0.041 (0.61–0.231)	0.138 ± 0.030 * (0.080–0.217)	0.119 ± 0.025 (0.072–0.167)	0.137 ± 0.036 ** (0.041–0.221)
Creatinine, mg/L	78.2 ± 26.8 (32.3–122.1)	99.5 ± 30.2 * (40.4–159.9)	78.0 ± 26.6 (43.2–139.0)	100.0 ± 23.6 ** (49.1–143.3)
Haemoglobin, g/L	105.9 ± 14.2 (67.0–131.0)	110.9 ± 12.2 (89.9–144.0)	108.4 ± 13.4 (86.0–145.0)	108.2 ± 13.5 (79.0–140.0)
RBC, ×10^12^/L	3.38 ± 0.48 (1.93–4.38)	3.48 ± 0.46 (2.76–4.46)	3.39 ± 0.46 (2.67–4.87)	3.37 ± 0.50 (2.45–4.59)
WBC, × 10^9^/L	7.00 ± 2.06 (4.01–14.26)	6.09 ± 2.39 * (3.00–11.80)	6.42 ± 1.90 (2.36–11.90)	6.14 ± 2.13 (3.00–12.30)
Albumin, g/L	43.5 ± 3.5 (36.6–57.2)	45.7 ± 4.8 (36.1–57.0)	41.1 ± 4.8 ** (33.2–51.3)	47.8 ± 5.1 ** (35.5–58.5)
Transferrin, g/L	2.41 ± 1.39 (0.54–6.07)	2.49 ± 1.54 (0.22–7.70)	2.23 ± 1.19 (0.35–6.84)	2.91 ± 1.58 ** (0.59–7.60)
Na, mmol/L	138.4 ± 2.23 (133–142)	138.9 ± 2.55 (133–143)	140.0 ± 2.26 (136–145)	137.5 ± 2.75 (132–142)
K, mmol/L	5.29 ± 0.67 (4.4–7.0)	5.40 ± 0.60 (4.3–7.2)	5.45 ± 0.68 (4.2–6.9)	5.60 ± 0.70 (4.5–7.0)
P, mmol/L	1.56 ± 0.55 (0.82–3.22)	1.83 ± 0.56 (0.89–2.94)	1.89 ± 1.35 (0.72–7.04)	1.90 ± 0.66 (0.75–3.10)
Ca, mmol/L	2.22 ± 0.20 (1.70–2.59)	2.21 ± 0.23 (1.51–2.52)	2.14 ± 0.23 (1.35–2.65)	2.30 ± 0.20 (1.76–2.59)

eGFR—estimated glomerular filtration rate, statistical significance at *p* < 0.05 * between HG1 and HG2 groups at the start of the study, ** in HG1 and HG2 groups at the end of the study (t = 24) with comparison with the basic value (t = 0).

**Table 3 biomolecules-13-00854-t003:** Putatively annotated metabolites for which signal intensity was significantly different in HG1 vs. HG2 at t = 24. ↑ ↓—changes in signal intensity when comparing HD1 vs. HD2.

*m*/*z*	Retention Time (min)	ESI Mode	*p*-Value	Name (ID HMDB)	Source/Pathway
230.0374	0.93	-	7.41 × 10^−10^ ↓	6-Hydroxymethyl-7,8-dihydropterin (HMDB0304227)	Food—pumpkin, spices
163.0397	0.54	+	9.10 × 10^−13^ ↑	4-Hydroxyphenylpyruvic acid (HMDB0000707)	Tyr, Phe decomposition metabolite
156.9961	1.28	-	8.04 × 10^−12^ ↓	Benzenesulphonic acid (HMDB0248986)	Xenobiotic
159.1023	0.76	-	6.76 × 10^−10^ ↑	5-Hydroxyvalproic acid (HMDB0013898)	Fatty acid metabolite
303.0175	0.52	-	4.32 × 10^−9^ ↓	p-Coumaric acid sulphate (HMDB0125166)	Food—whole grain
197,0927	0.70	+	1.45 × 10^−8^ ↑	4-(Glutamylamino)butanoic acid (HMDB0012161)	Polyamine, putrescine/orinitine, Arg catabolism
172.1087	3.07	+	6.26 × 10^−8^ ↑	Homocitrulline (HMDB0000679)	Alpha-amino acid, food—poultry
218.0455	3.62	-	4.28 × 10^−7^ ↑	2,6-Diamino-4-hydroxy-5-N-methylformamidopyrimidine (HMDB0011657)	Pyrimidine derivative—oxidative DNA damage
151.0259	2.68	-	5.13 × 10^−7^↓	6,8-Dihydroxypurine (HMDB0001182)	Purine nucleoside damaging DNA replication
146.0607	3.15	+	5.14 × 10^−7^↑	Indole-3-carboxaldehyde (HMDB0029737)	Trp catabolism, food—beans, cucumber, cereals and cereal products
207.1021	0.69	+	8.247 × 10^−7^ ↓	Diethyleneglycoldimethacrylate (HMDB0251246)	Derivative of dicarboxylic acid, xenobiotic
142.0869	1.79	+	8.51 × 10^−7^ ↑	Isovalerylglycine (HMDB0000678)	Acyl derivative of Gli metabolite—mitochondrial fatty acids, Leu catabolism
231.0771	1.63	+	1.93 × 10^−6^ ↑	5-Hydroxyindoleacetylglycine (HMDB0004185)	n-acyl-alpha-amino acid metabolite Trp, Val, Ile, Leu, food—poultry
142.1232	3.31	+	1.95 × 10^−6^ ↑	3-(Aminomethyl)-5-methylhexanoic acid (HMDB0245784)	Gamma-amino acid, xenobiotic
198.0404	2.05	-	3.05 × 10^−6^ ↑	5-(Hydroxymethyl-2-furoyl)glycine (HMDB0240476)	Alpha-amino acid, heat-treated food
128.0824	2.52	+	3.61 × 10^−6^↑	4-guanidinobutanoic acid (HMDB0003464)	Gamma-amino acid, Arg metabolite, food—fruit
248.0230	0.86	-	4.50 × 10^−6^ ↑	Norepinephrine sulphate (HMDB0002062)	Adrenaline/Tyr, Phe metabolite,
285.0433	0.64	-	7.38 × 10^−6^ ↑	Gamma-glutamylthreonine (HMDB0029159)	Protein ingestion/catabolism
290.0584	1.54	-	1.12 × 10^−5^ ↓	N(4)-acetylsulphisoxazole (HMDB0255283)	Xenobiotic
178.0872	1.98	-	1.43 × 10^−5^ ↑	Metanephrine (HMDB0004063)	Adrenaline/Tyr, Phe metabolite
206.0818	0.78	-	1.43 × 10^−5^ ↓	Tryptophol (HMDB0003447)	Indolyl alcohol, Trp catabolism

## Data Availability

The data presented in this study are available on request from corresponding authors. The data are not publicly available due to privacy and ethical restriction.

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
