# Peer review of "The Effects of Two Kinds of Dietary Interventions on Serum Metabolic Profiles in Haemodialysis Patients"

_biomolecules, 2023, doi:10.3390/biom13050854_

Round 1
Reviewer 1 Report
Nutritional status is an important factor for short-term and long-term outcomes in haemodialysis patients. At the moment there is not a single biomarker analysis that describes it, but there are several such as serum prealbumin, albumin, transferrin and creatinine.
This study aimed to evaluate the metabolic response to the administration of a meal immediately before the haemodialysis and to identify biomarkers of efficiency of dietary habits.
The findings of this study are consistent with previous studies that aimed to identify biomarkers useful for describing the nutritional status of haemodialyzed patients.
The Authors suggest that group are not homogeneous due to age, cause and duration of renal disease and comorbidities. They also claim that metabolomic profiling is only performed at the beginning and at the end of the study and that HG2 group patients do not always eat the entire meal before dialysis.
A further analysis of clinical data could tell if somehow metabolomic profiling is influenced by BMI, sex, physical activity and dietary habits of patients. A recent study by Annegrete Madsen et al. (Madsen A, Danielsen LMA, Niklassen AS, Hald M, Fjaeldstad AW, Bech JN, Ovesen T. Chemosensory function and food preferences among haemodialysis patients. Dan Med J. 2022 Sep 27;69(11):A08210644. PMID: 36331150.) suggests that chemosensory function is significantly poorer among haemodialyzed patients, so the authors recommend enhancement of odorant intensity and use of taste amplification in order to provide a better nutritional status.
Author Response
Dear Reviewer,
Thank you for comments to our manuscript. Response to your comments are in attachement. The changes are in red.
Your Comments: " They also claim that metabolomic profiling is only performed at the beginning and at the end of the study and that HG2 group patients do not always eat the entire meal before the dialysis"
Response: This sentence is not enough precise. It will be better t was incidential case that patients not always consume whole meal before HD.

Reviewer 2 Report
Comments and Suggestions for Authors
The manuscript is an original article that assessed the effects of dietary intervention on serum metabolic profile in haemodialysis (HD) patients. The topic is of great interest to nutritionists, clinicians, and basic researchers. The paper provides a generous introduction and sufficient background to understand its message, but requires improvement by reviewing a few major and minor issues:
Major
- In the introduction, the authors explained why HD patients have missing meals. There is plenty of evidence of the advantages of fasting (for example through the beneficial autophagy that occurs). Please explain and include in the manuscript why an extra meal versus a missing meal is more beneficial (knowing that in patients with chronic kidney disease is not recommended to load the kidney with protein metabolites).
- How did you evaluate if both groups of patients (HG1 and HG2) complied with the dietary recommendation? Perhaps the differences observed between the HG1 and HG2 groups are not necessarily related only to the meal taken before HD in the HG2 group. Wouldn't it have been more accurate if you had compared the same patients (not two different groups) at 24 weeks with and without a meal before HD to have effective comparability? Please insert a comment in the manuscript on this aspect related to the design of the study.
- The p-value is missing in Table 1 to see if there are significant differences between the two groups.
- In the discussion chapter, explain what would be the applicability of discovering the differences between the HG1 and HG2 groups in terms of amino acid metabolites. Or if it would have diagnostic or prognostic value, taking into account that the majority of amino acids were essential (normally brought through the diet) and not non-essential.
- The fact that HG2 patients did not always consume the entire meal before HD could lead to a bias in the data obtained when compared to HG1 patients. Explain how this could influence the statistical data and how you minimized this bias.
- Please explain in the discussion chapter why the study only evaluated metabolites of amino acids, fatty acids, and pyrimidines metabolism, and no metabolites of carbohydrate metabolism.
Minor
1. There are some phrases in the introduction which do not make sense (probably words or verbs are missing). Examples: “Persistent depletion of proteins and energy in the body due to poor nutritional status, caused by multiple nutritional and non-nutrition mechanisms, as well as metabolic alter-ations (Pauzi et al., 2020)” (probably “is caused”); “No direct evidence has been evaluated of metabolic response to the HD procedure with a meal administered immediately before the haemodialysis” (probably “for the metabolic response”); etc.
2. The authors did not put the references in square brackets, as required by the journal.
3. Throughout the text, the authors were not consistent regarding the abbreviations. Example: haemodialysis, abbreviated as HD, was used unabbreviated in many places.
After the amendment of the above comments in the manuscript, I would be in favor of publishing this paper.
Author Response
Dear Reviewer,
Thank you for reviewing the manuscript.Please see the attachment. In attachment is response or explanation to your comments and corrected manuscript with changes in red.

Round 2
Reviewer 2 Report
Comments and Suggestions for Authors
The manuscript is an original article that assessed the effects of dietary intervention on serum metabolic profile in haemodialysis (HD) patients. I am glad that the authors made essential improvements to the manuscript.
In this sense, they responded adequately to the 6 major and 3 minor points that I addressed.
Also, clarifications were made in the discussions and conclusions chapters.
In conclusion, the changes made to the article are significant and can support its publication in the journal.
Author Response
Dear Reviewer,
Thank you for your opinion on revised manuscript. Thank you for all important comment to our manuscript.